# Trigeminal Nerve Affection in Patients with Neuro-Sjögren Detected by Corneal Confocal Microscopy

**DOI:** 10.3390/jcm11154484

**Published:** 2022-08-01

**Authors:** Tabea Seeliger, Marten A. Gehlhaar, Irene Oluwatoba-Popoola, Franz F. Konen, Melanie Haar, Emilia Donicova, Marija Wachsmann, Amelie Pielen, Stefan Gingele, Nils K. Prenzler, Diana Ernst, Torsten Witte, Carsten Framme, Anna Bajor, Thomas Skripuletz

**Affiliations:** 1Department of Neurology, Hannover Medical School, 30625 Hannover, Germany; seeliger.tabea@mh-hannover.de (T.S.); konen.felix@mh-hannover.de (F.F.K.); gingele.stefan@mh-hannover.de (S.G.); 2Department of Ophthalmology, Hannover Medical School, 30625 Hannover, Germany; gehlhaar.marten@mh-hannover.de (M.A.G.); oluwatoba-popoola.irene@mh-hannover.de (I.O.-P.); haar.melanie@mh-hannover.de (M.H.); emilia.donicova@gmail.com (E.D.); wachsmann.marija@mh-hannover.de (M.W.); pielen.amelie@mh-hannover.de (A.P.); framme.carsten@mh-hannover.de (C.F.); bajor.anna@mh-hannover.de (A.B.); 3Department of Otolaryngology, Hannover Medical School, 30625 Hannover, Germany; prenzler.nils@mh-hannover.de; 4Department of Rheumatology & Immunology, Hannover Medical School, 30625 Hannover, Germany; ernst.diana@mh-hannover.de (D.E.); witte.torsten@mh-hannover.de (T.W.)

**Keywords:** corneal confocal microscopy, corneal nerve fibre density, Neuro-Sjögren, Sjögren’s syndrome associated neuropathy

## Abstract

Background: Patients with Sjögren’s syndrome and polyneuropathy more frequently develop cranial nerve affection when compared to patients with chronic inflammatory demyelinating polyneuropathy (CIDP). We therefore aimed to analyze trigeminal corneal nerve fibre characteristics in both patient groups. Methods: A total of 26 patients with Sjögren’s syndrome associated neuropathy and 29 patients with CIDP were recruited at our university hospital and compared to 6 healthy controls. Dry eye symptoms and signs were assessed via clinical examination and the Ocular Disease Surface Index questionnaire. Trigeminal corneal nerve fibres were analyzed via corneal confocal microscopy (CCM) as a non-invasive in vivo microscopy. Results: CCM revealed significantly reduced corneal nerve fibre density and corneal nerve fibre main branch density in the Neuro-Sjögren group when compared with healthy controls. There were no significant group differences between the Neuro-Sjögren and the CIDP group for any of the microscopic parameters. Dry eye assessment showed similarly reduced scores for both patient groups, while healthy controls showed better results for objective dry eye signs. There was no correlation between microscopic parameters of the corneal confocal microscopy and parameters of dry eye assessment. Conclusions: Our data revealed trigeminal corneal nerve affection in patients with neuropathy associated with Sjögren’s syndrome and patients with CIDP detected by CCM. No difference was found between both neuropathy groups indicating that CCM is not able to distinguish between both entities.

## 1. Introduction

Sjögren’s syndrome has become increasingly recognized for its neurological involvement in recent years [1,2,3,4], forming the entity of Neuro-Sjögren. Peripheral nervous system impairment is most commonly found in affected patients, but the disease can also manifest at the central nervous system or at muscular structures in the sense of myositis [5,6,7]. Still, many clinical characteristics of Neuro-Sjögren are yet to be investigated.

Previous studies have found many similarities between the neuropathy associated with Sjögren’s syndrome and other autoimmune-mediated neuropathies such as chronic inflammatory demyelinating polyneuropathy. Both entities are associated with, for example, focal thickening of peripheral nerves and/or prominent fascicles [8]. In a comparison of CIDP patients with and without Sjögren’s syndrome, only cranial nerve impairment and female sex were more common in the subgroup with additional Sjögren’s syndrome, while clinical presentation as well as electrophysiological and laboratory findings of cerebrospinal fluid were similar [9].

The cornea is the most densely innervated tissue in the human body, which accounts for its extreme sensitivity. It measures about 12 mm in diameter, has a thickness of about 0.5 mm centrally and consists of five distinct layers namely: the epithelium, Bowman’s layer, the stroma, Descemet’s membrane and the endothelium. Its nerve fibres originate from the ophthalmic division of the trigeminal nerve, which eventually terminates in branched nerve fibres that form the subbasal nerve plexus (SBNP) located between the epithelium and Bowman’s layer [10]. Among the components of this avascular tissue, the nerves are of particular interest, as changes in their morphology and/or quantification serve as pointers towards understanding the presentation and progression of a wide range of not only keratopathies, but also neurovascular and neurodegenerative diseases. The characteristics of corneal nerves in systemic conditions associated with peripheral neuropathies such as diabetes, Fabry’s disease, Behçet’s and Parkinson’s disease have been studied [11,12,13,14,15,16].

Corneal confocal microscopy is increasingly becoming a valuable non-invasive tool in both diagnostics and research, by which corneal structure in systemic medical conditions can be examined on a cellular level [17,18]. Most importantly, this method has already been shown to reliably detect trigeminal nerve fibre damage in patients with CIDP [19,20,21].

We therefore aimed to characterize the corneal trigeminal nerve affection in patients with Sjögren’s syndrome associated neuropathy in comparison to patients with chronic inflammatory demyelinating polyneuropathy (CIDP) and to evaluate possible connections with dry eye disease symptoms and signs.

## 2. Materials and Methods

### 2.1. Study Design

Ophthalmological examination by corneal confocal microscopy as well as clinical evaluation of dry eye symptoms and signs were prospectively performed among three groups: patients with Sjögren’s syndrome and clinical impairment of peripheral nerves (Neuro-Sjögren), patients with CIDP and a control group of healthy volunteers. Data were analyzed for in-between group differences and correlation between parameters of the corneal confocal microscopy and the clinical evaluation of dry eye symptoms and signs.

Patients were recruited between 04/2019 and 09/2021 at the Hannover Medical School (departments of neurology and ophthalmology) from both outpatient clinics and wards. Written informed consent to study participation and data analysis was obligatory. Exclusion criteria were ophthalmological conditions technically interfering with the corneal confocal microscopy (such as corneal ulcers and recently performed ophthalmological invasive procedures) and advanced neurological invalidity that would prohibit participants from complementing the laborious testing. Additionally, patients with topical ophthalmic treatment other than tear substitutes were not included. 

### 2.2. Diagnosis of Sjögren’s Syndrome and Evaluation of Disease and Symptom Severity

Patients in the Neuro-Sjögren group had to fulfill the current classification criteria of the American College of Rheumatology/European League Against Rheumatism (ACR/EULAR) as previously described [4,22]. Every patient was therefore evaluated for the established Focus Score derived by summation of the following items: objective xerophthalmia (1 point) and xerostomia (1 point) on formal testing via Schirmer’s test and Saxon test, respectively; Anti-SSA/Ro-antibody-positivity (3 points) and sialadenitis with ≥one evident lymphocytic focus/mm^2^ (3 points) [23]. Sjögren’s syndrome was diagnosed if the Focus Score resulted in values ≥4.

For patients with Sjögren’s syndrome, disease severity was surveyed by the EULAR Sjögren’s Syndrome Disease Activity Index (ESSDAI), while symptom severity concerning Sjögren’s syndrome was evaluated by the EULAR Sjögren’s Syndrome Patient Report Index (ESSPRI) [24]. The ESSDAI evaluates organ manifestations of Sjögren’s syndrome through domain subscores: “constitutional”, “lymphadenopathy”, glandular”, “articular”, “cutaneous”, “pulmonary”, “renal”, “muscular”, “peripheral nervous system”, “central nervous system”, “haematological” and “biological”. The cumulative score ranges from 0 (no organ involvement) to 123 (maximum involvement in all domains). The ESSPRI comprises of three items (dryness, fatigue and pain) and ranges from 0 (no complaints) to 10 (maximal complaints). Patients of the CIDP group had to fulfill the current diagnostic criteria of the European Federation of Neurological Societies/Peripheral Nerve Society (EFNS/PNS) [25] while diagnostic workup of Sjögren’s syndrome had to be unambiguously negative. Serological biomarkers of the CIDP group were not evaluated [26].

### 2.3. Corneal Confocal Microscopy

The corneal confocal microscopy is a highly resolving laser microscopic technique that allows an accurate, non-invasive assessment of the corneal nerve fibres in vivo [27] and was conducted by experienced investigators for this study. Microscopic corneal image acquisition was carried out using the Rostock Cornea Module (RCM) mounted on the HRT III tomograph (Heidelberg Engineering GmbH, Heidelberg, Germany, while using the recommended resolution of 400/384 = 1.0417 µm. Each eye was first anaesthetized using a drop of Proparacaine (Proxymetacaine Hydrochloride 0.5%, Ursapharm Arzneimittel GmbH, Saarbrücken, Germany) followed by the instillation of a gel tear substitute (Vidisic Gel, Carbomer 2 mg per 1 g, Bausch and Lomb, Brunsbuetteler Damm 165/173, Berlin, Germany) which served as a protective layer over the corneal epithelium layer. A sterile TomoCap (Heidelberg Engineering GmbH) was placed over the RCM objective lens before contact was made with the anaesthetized corneal surface.

Image analysis was conducted by the ACC Metrics Software© V1 2015, Manchester, UK.

The following parameters were automatically calculated:Corneal Nerve Fibre Density (CNFD): The number of detected fibres per mm^2^Corneal Nerve Fibre Main Branch Density (CNBD): The number of detected branch points on the main fibres per mm^2^Corneal Nerve Fibre Length (CNFL): The total length of detected nerve fibres in mm per mm^2^Corneal Nerve Fibre Total Branch Density (CTBD): The total number of branch points per mm^2^

### 2.4. Clinical Evaluation of Dry Eye Symptoms and Signs

Dry eye symptoms and signs were evaluated by general ophthalmological examination quantified by the predefined “clinical score of the dry eye” and the established Ocular Surface Disease Index Questionnaire (OSDI):

#### 2.4.1. Clinical Score of the Dry Eye

Each eye underwent a comprehensive examination of the anterior segment of the eye with the Slit lamp, conducted by experienced ophthalmologists. Ocular surface integrity was assessed with the cobalt blue filter following fluorescein staining with a drop of Thilorbin (4 mg Oxybuprocaine Hydrochloride + 0.8 mg Fluorescein Sodium), OmniVision GmbH. The ophthalmological examination was quantified by the “Clinical score of the dry eye”, a modified version of existing recommendations for dry eye assessment [28,29,30]. This score comprises evaluation of conjunctival injection, tear film break-up time, blink frequency [31,32], the presence and extent of superficial punctate keratitis and the presence of meibomian gland dysfunction. The scoring system is shown in Table 1. Eventually, the sum score for each eye ranged from 0 (no evidence of dry eye disease) to 13 (severe manifestation in all aspects).

#### 2.4.2. Ocular Surface Disease Index (OSDI)

The dry eye associated symptoms with impact on the quality of daily life were evaluated by the OSDI [36,37]. This index consists of 12 questions concerning ocular symptoms, vision-related function and environmental triggers during the previous week, while scores range from 0 (none of the time) to 4 (all of the time), respectively. The total OSDI sum score is subsequently formulated through the sum of scores × 25 divided by the number of answered questions. It therefore ranges from 0 (no complaints) to 100 (all symptoms all the time). 

### 2.5. Statistical Analysis and Data Visualization

Decimal variables were tested for parametrical distribution via the Shapiro–Wilk test. Parametrical data were described as mean ± standard deviation, whereas non-parametrical data were described as median (interquartile range). After descriptive analysis as designated, group comparison was achieved via the Wilcoxon Rank sum Test for decimal data and via Chi² test for binary data. Correlation analysis was performed using Spearman’s rank correlation. *p*-values < 0.05 were considered significant. Statistical analysis was performed by STATA^®^ V16.1 (StataCorp LLC, 4905 Lakeway Dr, College Station, TX, USA). Figures were created by RStudio Desktop V1.4.1717 (RStudio Inc, 250 Northern Ave, Boston, MA, USA).

## 3. Results

### 3.1. Patients’ Characteristics

A total of 26 patients with Sjögren’s syndrome (according to the current ACR/EULAR classification criteria) and evident neuropathy (Neuro-Sjögren), as well as 29 patients with previously diagnosed chronic inflammatory demyelinating polyneuropathy (CIDP) and without Sjögren’s syndrome were included in the analysis. Additionally, six control patients without Sjögren’s syndrome or CIDP underwent identical testing. Patients in the Neuro-Sjögren group were female in 58% and showed a median age at evaluation of 64 years, while patients in the CIDP group were female in 21% and showed a median age of 67 years. Controls were female in 83% and aged 61 years (median). The cohort characteristics concerning the ACR/EULAR classification criteria of Sjögren’s syndrome, the disease duration, as well as disease and symptom severity are shown in detail in Table 2.

The serological and clinical characteristics of patients with neuropathy associated with Sjögren’s syndrome are displayed in Table 3.

### 3.2. Confocal Microscopy of the Corneal Nerve Fibres

Confocal microscopy of the corneal nerve fibres revealed no significant group differences between the Neuro-Sjögren and the CIDP group for any of the microscopic parameters (values represent mean ± standard deviation of median (interquartile range) as appropriate): 

Corneal Nerve Fibre Density (fibres/mm^2^): Neuro-Sjögren group 17.1 ± 6.8 (left) and 16.9 ± 8 (right); CIDP group 15.9 ± 7.7 (left, *p* = 0.27) and 15.2 ± 6.8 (right, *p* = 0.29).

Corneal Nerve Fibre Main Branch Density (main branch points/mm^2^): Neuro-Sjögren group 17.2 (9.4–27.1) (left) and 15.6 (3.1–26.6) (right); CIDP group 15 (6.3–24.3) (left, *p* = 0.6) and 15.3 (6.3–25) (right, *p* = 0.68).

Corneal Nerve Fibre Length (total length of nerves) (mm/mm^2^): Neuro-Sjögren group 10.6 ± 3.9 (left) and 12.1 (6.2–14.1) (right); CIDP group 10.9 ± 3.6 (left, *p* = 0.93) and 10 (7–13.3) (right, *p* = 0.73).

Corneal Nerve Fibre Total Branch Density [total branch points/mm^2^]: Neuro-Sjögren group 33.3 (17.7–41.4) (left) and 25.2 (10.9–40.6) (right); CIDP group 26.3 (16.5–43,8) (left, *p* = 0.88) and 25 (16.7–43.6) (right, *p* = 0.67).

Control participants showed significantly better results with the corneal nerve fibre density (24.8 ± 5.8/mm^2^, *p* = 0.02 (left) and 28.9 ± 5.4/mm^2^, *p* = 0.002 (right)) and the corneal nerve fibre length (total length of nerves) (14.6 ± 1.5 mm/mm^2^, *p* = 0.004 (left) and 14.6 (13.6–16.6) mm/mm^2^, *p* = 0.02 (right). For the corneal nerve fibre main branch density (when compared to the Neuro-Sjögren group), significantly better results were evident only in the right eyes (23.1 (18.7–41.2)/mm^2^, *p* = 0.15 (left) and 30 (27.5–32.5), *p* = 0.02 (right)). Analysis of the corneal nerve fibre total branch density did not reveal significant group differences between patients with Neuro-Sjögren and controls (36.4 (25–68.7), *p* = 0.3 (left) and 39.4 (28.7–45), *p* = 0.15 (right)).

The detailed workup for the confocal microscopy results is illustrated in Figure 1. Exemplary findings in Neuro-Sjögren patients and rarefied corneal nerve fibres versus control participants are shown in Figure 2.

### 3.3. Clinical Evaluation of Dry Eye Symptoms and Signs

Considering the clinical score of the dry eye and the Ocular Surface Disease Index, there were no significant in-between-group differences between patients with Neuro-Sjögren and CIDP. Nevertheless, controls reached significantly better results at the clinical score of the dry eye when compared with the Neuro-Sjögren group (values are stated as median (interquartile range)): Neuro-Sjögren group 6 (4–8) (left) and 6 (4–8) [right]; CIDP group 4 (3–5) (left, *p* = 0.12) and 4 (3–6) (right, *p* = 0.12); controls 1 (0–3) (left, *p* = 0.003) and 2 (1–3) (right, *p* = 0.003).

However, the OSDI score was similar for all three groups (values are stated as median (interquartile range)): Neuro-Sjögren group 16.4 (6.8–35); CIDP group 10.4 (2.1–20.8) (*p* = 0.26), controls 12.5 (4.2–16.7 (*p* = 0.38).

The results of the Clinical evaluation of dry eye symptoms and signs are graphically demonstrated in Figure 3.

### 3.4. Correlation Analysis between Parameters of the Corneal Confocal Microscopy and the Clinical Evaluation of Dry Eye Symptoms and Signs 

There was no significant correlation between any of the parameters of the corneal confocal microscopy and the final sum score results of the Clinical score of the dry eye or the Ocular Surface Disease Index (Table 4). Nevertheless, there was a significant correlation between the clinical score of the dry eye and the Ocular Surface Disease Index (*p* = 0.005 (left) and *p* = 0.002 (right)). 

## 4. Discussion

### 4.1. Corneal Trigeminal Nerve Affection Is Similar in Neuro-Sjögren and CIDP 

Analysis of the corneal nerve parameters on corneal confocal microscopy did not show significant group differences between patients with Neuro-Sjögren and patients with CIDP. This finding aligns with previously published data suggesting a similar phenotype of these two entities, even though cranial nerve affection had been found to occur more frequently in patients with Neuro-Sjögren [9].

Although there are no concretely established normal values or cutoff recommendations for the parameters of the corneal confocal microscopy, efforts have been made towards establishing reference values against which corneal confocal microscopy findings can be measured [38]. In this study, the absolute values of the corneal nerve fibre density and the corneal nerve fibre length were significantly reduced in patients with Neuro-Sjögren when compared to healthy controls. The same effect has been previously described for patients with Behçet’s disease [39], non-Sjögren dry eye disease [40], Sjögren’s syndrome without focus on neurological involvement [41,42,43,44,45], diabetes [43], hereditary [46] and autoimmune neuropathies other than Sjögren’s syndrome associated neuropathy [19,47]. Interestingly, the published absolute values for corneal nerve fibre density and length vary depending on the disease. A literature review of comparable measurements is included in Table 5. Still, there is a slight variability previously published on the respective values for control patients and the values for the right and left eye also differing slightly within our control group. This effect might be explained by differences for age and size of the control cohorts, but possibly also by the technical nature of the testing procedure. For example, it is essential for patients to remain very still in order to obtain representative corneal images, which might be difficult in case of additionally limiting physical disabilities or conditions such as general or truncal muscle weakness or disrupted oculomotor functions. Nevertheless, the automated nature of image analysis through the provided ACC Metrics Software© offers a standardized data assessment with great validity.

It is noteworthy that only one male patient was included in the control group to align the ratio of women and men for the control group and for patients with Sjögren’s syndrome. However, no sex-specific influence on corneal confocal microscopy could be found in previous works [38,48]. Therefore, a resulting relevant confounding effect seems unlikely.

### 4.2. Dry eye Disease in Both Patient Groups

Clinical evaluation of dry eye symptoms and signs by ophthalmological examination showed significantly worse results for patients with Neuro-Sjögren and CIDP when compared to control participants. However, results of the OSDI questionnaire did not differ between the three groups due to a great variation of subjective suffering in all groups. We deduced that the clinical examination is more precise in quantifying dry eye symptoms and signs than the OSDI, which also includes general aspects (i.e., having eye problems that limit TV watching) possibly affected by confounding factors (i.e., refraction anomalies etc.). 

Nevertheless, there was a significant correlation between scores of the clinical examination and the OSDI supporting a general connection between the objective and subjective evaluation of dry eye symptoms and signs as reported by previous studies [40].

Another aspect of these results is that patients with Neuro-Sjögren and patients with CIDP had similar ophthalmological findings concerning signs of dry eye disease. This supports the fact that dry eye disease is by no means a specific characteristic of Sjögren’s syndrome, even though lymphocytic infiltration of the lacrimal glands is a known hallmark for this entity. On the contrary, sicca symptoms have been previously described in chronical diseases [50,51]. It is also possible that neurological involvement of corneal nerve fibres in patients with Sjögren’s syndrome precedes more severe sicca symptoms as has been shown for other neurological manifestations of Sjögren’s syndrome [52,53].

### 4.3. Notional Connection between Dry Eye Disease and Corneal Trigeminal Nerve Affection

Both patient groups showed a similar severity of clinical ophthalmological sicca syndrome and corneal nerve fibre affection. Still, parameters of the corneal confocal microscopy did not show a significant correlation with either of the clinical scores. Therefore, a potential reciprocal impact of both conditions remains notional. 

## 5. Conclusions

Our data revealed trigeminal nerve affection at the cornea in patients with neuropathy associated with Sjögren’s syndrome and patients with CIDP. No difference was found between both neuropathy groups, indicating that measurement of corneal nerve fibre/branch density and fibre length is not able to distinguish between both entities. 

## Figures and Tables

**Figure 1 jcm-11-04484-f001:**
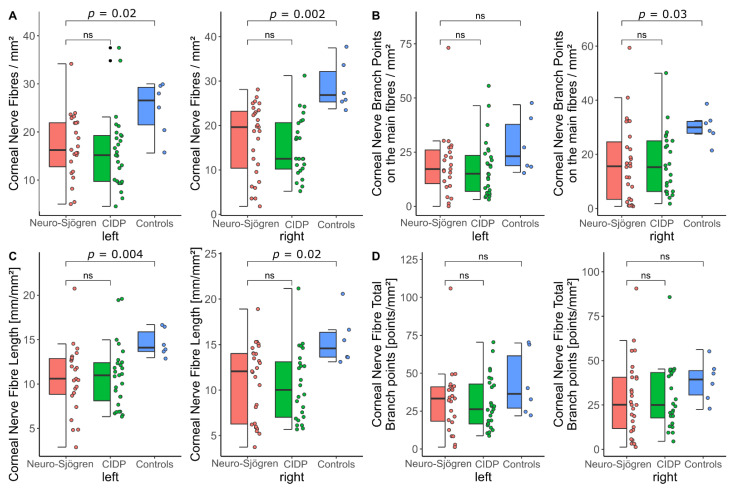
Confocal microscopy parameters with significance levels at in-between group differences for patients with Neuro-Sjögren (red), chronic inflammatory demyelinating polyneuropathy (CIDP, green) and controls (blue): (**A**) corneal nerve fibre density; (**B**) corneal nerve fibre main branch density; (**C**) corneal nerve fibre length (total length of nerves); (**D**) corneal nerve fibre total branch density; ns: not significant.

**Figure 2 jcm-11-04484-f002:**
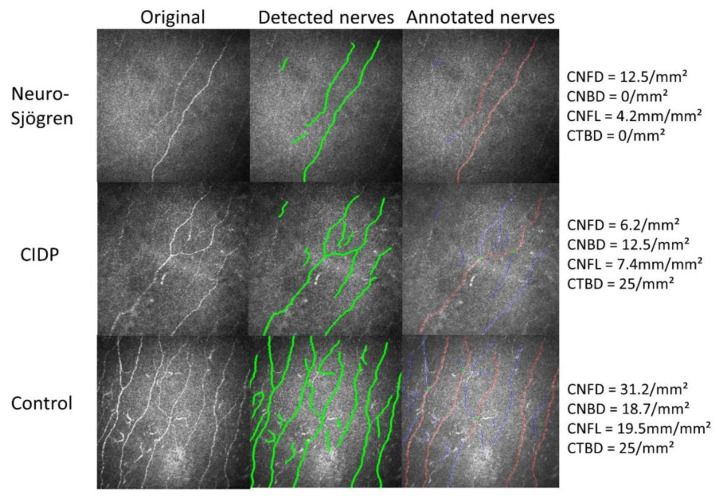
Exemplary findings in one patient with Neuro-Sjögren, one with CIDP and one participant of the control group. Image analysis is displayed for the automatically detected and colored nerves (green) and the annotated nerve fibres (red: main fibres, blue: nerve branches, green dots–branch points at the main fibres). Abbreviations; CIDP: chronic inflammatory demyelinating polyneuropathy; CNFD: corneal nerve fibre density; CNBD: corneal nerve fibre main branch density; CNFL: corneal nerve fibre length (total length of nerves); CTBD: corneal nerve fibre total branch density.

**Figure 3 jcm-11-04484-f003:**
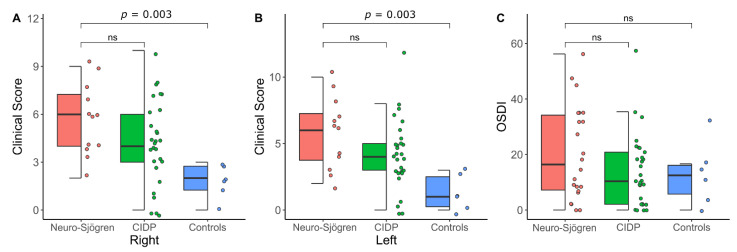
Results of the Clinical Evaluation of dry eye symptoms and signs with significance levels at in-between group differences for patients with Neuro-Sjögren (red), chronic inflammatory demyelinating polyneuropathy (CIDP, green) and controls (blue): clinical score of the dry eye for the right (**A**) and left side (**B**); ocular surface disease index (OSDI, **C**); ns: not significant.

**Table 1 jcm-11-04484-t001:** Clinical score of the dry eye, applied for each eye separately *, grey color indicating non-applicable cells.

	0	1	2	3
Conjunctival Injection [33]	none	mild	moderate	severe
Tear Film Breakup Time	>10 s	≤10 s	≤5 s	immediate
Blink Frequency	≤21/min	>21/min		
Schirmer’s test (mm/5 min)	>10	≤10	≤5	≤2
Superficial Punctate Keratitis (National Eye Institute Score) [34]	Grade 0	Grade 1	Grade 2	Grade 3
Meibomian Gland Dysfunction [35]	none	mild	moderate	severe

* Modified version of existing recommendations for dry eye assessment [28,29,30].

**Table 2 jcm-11-04484-t002:** Baseline parameters for the analyzed subgroups: Neuro-Sjögren, chronic inflammatory demyelinating polyneuropathy (CIDP) and controls.

	Neuro-Sjögren	CIDP	Controls
*n*	26	29	6
Age at evaluation, median (IQR) (years)	64 (56–72)	67 (60–74)	61 (60–64)
Female, *n* (%)	15 (58%)	6 (21%)	5 (83%)
Disease duration, median (IQR) (years)	7 (2–18)	9 (4–10)	n/a
ACR/ EULAR classification criteria			
Objective xerophthalmia, *n* (%)	20 (77%)	7 (24%)	n/a
Objective xerostomia, *n* (%)	12 (46%)	8 (28%)	n/a
SSA(Ro) antibody positivity, *n* (%)	11 (42%)	0 (0%)	n/a
Sialadenitis grade 3/4 by Chisholm and Mason, *n* (%)	21/22 (95%)	0/10 (0%)	n/a
Focus Score, median (IQR)	4 (4–5)	0 (0–1)	n/a
ESSDAI, median (IQR)	17 (11–22)	n/a	n/a
ESSPRI, median (IQR)	3.2 (2.3–4.8)	n/a	n/a

Abbreviations: CIDP: chronic inflammatory demyelinating polyneuropathy; *n*: number of patients; IQR: interquartile range; ACR/EULAR: American College of Rheumatology/European League Against Rheumatism; n/a: not applicable.

**Table 3 jcm-11-04484-t003:** Additional serological and clinical features of the subgroup of patients with Sjögren’s syndrome and associated neuropathy.

	*n* (%)
**Clinical feature of peripheral nerve affection**	
Cranial nerve impairment	9 (35%)
Small Fibre Neuropathy	2 (8%)
Motor impairment	21 (81%)
Sensory deficits	21 (81%)
● Pain	4 (19%)
● Paresthesia	12 (57%)
● Sensory ataxia	13 (62%)
Autonomic dysfunction	4 (16%)
**Nerve damage pattern on pathological nerve conduction studies (*n* = 24)**	
SSB(La)-antibody positivity	3 (12%)
Cryoglobulins	0 (0%)
**Additional serological parameters**	
SSB(La)-antibody positivity	3 (12%)
Cryoglobulins	0 (0%)
Free Kappa Light Chains, median (IQR) (mg/L)	19 (21.1–21.5)
Free Lambda Light Chains, median (IQR) (mg/L)	19.7 (11.4–22.3)
Kappa/Lambda ratio, median (IQR)	(0.81–1.21)

**Table 4 jcm-11-04484-t004:** Correlation analysis for the full cohort between parameters of the corneal confocal microscopy and the clinical evaluation of dry eye symptoms and signs with significance levels (green color-significant group differences).

Analysis without Subgroup Division	Left	Right
Clinical score, mean ± standard deviation	4 (3–6.5)	4 (3–6.5)
OSDI	12.5 (4.2–25)	
Correlation analysis	*p*-values (left)	*p*-values (right)
Clinical Score vs. OSDI	0.005	0.002
Clinical Score vs.		
Corneal Nerve Fibre Density	0.34	0.6
Corneal Nerve Fibre Main Branch Density	0.44	0.88
Corneal Nerve Fibre Length (total length of nerves)	0.13	0.41
Corneal Nerve Fibre Total Branch Density	0.66	0.74
OSDI vs.		
Corneal Nerve Fibre Density	0.50	0.68
Corneal Nerve Fibre Main Branch Density	0.40	0.48
Corneal Nerve Fibre Length (total length of nerves)	0.14	0.42
Corneal Nerve Fibre Total Branch Density	0.08	0.30

**Table 5 jcm-11-04484-t005:** Literature review on published data concerning corneal confocal microscopy parameters in different cohorts.

Reference	Patients				Controls			
	Patients’ Condition, *n*	Median Age	Corneal Nerve Fibre Density, Mean ± SD (fibres/mm^2^)	Corneal Nerve Fibre Length, Mean ± SD (mm/mm^2^)	*n*	Median Age	Corneal Nerve Fibre Density, Mean ± SD (fibres/mm^2^)	Corneal Nerve Fibre Length, Mean ± SD (mm/mm^2^)
[39]	Behçet’s disease, 49	39.9	27.7 ± 8.6	16.3 ± 4.6	30	41.2	35.6 ± 10	18.5 ± 4.1
[40]	Dry eye disease, 43	46.2	34.9 ± 8.1	16.26 ± 3.5	14	45.4	45.9 ± 4.2	21.86 ± 2.1
[42]	Sjögren’s syndrome, 10	58.2	21.7 ± 18.9	4.18 ± 3.4	10	56.5	31.8 ± 9.3	6.54 ± 2.47
[49]	Sjögren’s syndrome with dry eye disease, 54	57.8	28.1 ± 12.2	10.3 ± 6.6	20	50.9	43.9 ± 12.9	15.4 ± 5.1
[43]	Diabetes, 998	52	20.6 ± 9.8	12.5 ± 4.6	-	-	-	-
[19]	CIDP, 88	n/a	19 ± 7	12 ± 3	85	n/a	29 ± 6	17 ± 3
	MMN, 6	n/a	18 ± 11	11 ± 5				
	MGUSN, 12	n/a	20 ± 5	12 ± 2				

## Data Availability

Data are available upon reasonable request.

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
