# Peer review of "Trigeminal Nerve Affection in Patients with Neuro-Sjögren Detected by Corneal Confocal Microscopy"

_jcm, 2022, doi:10.3390/jcm11154484_

Round 1
Reviewer 1 Report
1. Please explain the meaning of grey color in table 1.
2. It is weird to see references [20-22] in title of table1.
3. Please explain the meaning of green color in table 3.
4. The authors should explain the sentences “published absolute values for both parameters show a slight variability, which 264 might be explained by age- and cohort-size related effects, but also by the challenging 265 technical nature of the testing procedure: For example, it is essential for patients to remain 266 very still in order to obtain representative corneal images, which might be difficult in case 267 of physical disabilities.” more detailed.
5. The authors should explain why they use * for p<0.05; ** for p<0.01 in figure 1 and figure 3. Is it necessary?
Author Response
First of all, we would like to thank the kind reviewers for taking time to review our manuscript and providing a fair and critical feedback. We have addressed all points raised by the reviewers and provide a detailed point to point response.
Comment 1: Please explain the meaning of grey color in table 1.
Response 1: The grey color indicates non-applicable cells. We included this information in the legend of table 1.
Comment 2: It is weird to see references [20-22] in title of table1.
Response 2: We agree and relocated the information on the references to the footnote of table 1.
Comment 3: Please explain the meaning of green color in table 3.
Response 3: The green color indicates significant group differences. We included this information in the legend of table 3.
Comment 4: The authors should explain the sentences “published absolute values for both parameters show a slight variability, which 264 might be explained by age- and cohort-size related effects, but also by the challenging 265 technical nature of the testing procedure: For example, it is essential for patients to remain 266 very still in order to obtain representative corneal images, which might be difficult in case 267 of physical disabilities.” more detailed.
Response 4: We apologize for the misleading description and rephrased the according paragraph as follows:
Interestingly, the published absolute values for corneal nerve fibre density and length vary depending on the disease. A literature review of comparable measurements is included in table 5. Still, there is a slight variability previously published respective values for control patients and the values for the right and left eye also differed slightly within our control group. This effect might be explained by differences for age and size of the control cohorts, but possibly also by the technical nature of the testing procedure: For example, it is essential for patients to remain very still in order to obtain representative corneal images, which might be difficult in case of additionally limiting physical disabilities or conditions such as general or truncal muscle weakness or disrupted oculomotor functions.
Comment 5: The authors should explain why they use * for p<0.05; ** for p<0.01 in figure 1 and figure 3. Is it necessary?
Response 5: It is true that the p-value differentiation between the cutoffs of 0.05 and 0.01 does not help with clarification of the results. We therefore chose to include the exact p-values into the figure for significant p-values.
Reviewer 2 Report
Dear authors,
The study of corneal confocal microscopy is of great interest not only to ophthalmology but to other subspecialties, so congratulations on the theme of your work.
I have a few suggestions in order to improve your manuscript.
In the introduction, there is a lack of explanation of what characterizes CIDP and why you chose to compare this particular disease with Neuro-Sjogren. I feel like the background of your abstract states information that is missing in the introduction itself. Specifically in line 52, change "an invaluable" to "a valuable".
The discussion lacks the important limitation of your work of having only six controls, a reduced number when compared to the neuro-sjogren and CIDP group. This could explain why disparities were found between right and left eye of controls in the analysis of Corneal Nerve Fibre Main Branch Density.
It is also important to note that there was only one male control. Does literature state any difference in measurements of CCM between genders? If not, state this in the discussion.
The conclusion correctly correlates to the aim of your work, both mentioning the comparison of Neuro-Sjogren with CIDP, but this is not made clear in the title of your work. Therefore, I believe it to be necessary for you to either better justify this choice of comparison - is CCM already established as a diagnostic tool in CIDP? - or to change your title to better describe the scope of your work.
Hope to have made valuable contributions to your work.
Author Response
First of all, we would like to thank the kind reviewers for taking time to review our manuscript and providing a fair and critical feedback. We have addressed all points raised by the reviewers and provide a detailed point to point response.
Comment 1: In the introduction, there is a lack of explanation of what characterizes CIDP and why you chose to compare this particular disease with Neuro-Sjogren. I feel like the background of your abstract states information that is missing in the introduction itself.
Response 1: We thank this reviewer for the positive feedback and are grateful for pointing out this shortcoming in the introduction. A new second paragraph and an additional explanation at the end of the discussion has been implemented to fill in the missing information.
Comment 2: Specifically in line 52, change "an invaluable" to "a valuable".
Response 2: Changed accordingly.
Comment 3: The discussion lacks the important limitation of your work of having only six controls, a reduced number when compared to the neuro-sjogren and CIDP group. This could explain why disparities were found between right and left eye of controls in the analysis of Corneal Nerve Fibre Main Branch Density.
Response 3: This is an important point, which is now included in the discussion section.
Comment 4: It is also important to note that there was only one male control. Does literature state any difference in measurements of CCM between genders? If not, state this in the discussion.
Response 4: It is true, that we only included one male patient in the control group in order to align the female/male ratio of controls and patients with Sjögren’s syndrome. Interestingly, previous studies have not shown a gender-related influence on corneal confocal microscopy parameters. We therefore deduced that gender is unlikely to cause a relevant confounding effect, but we still included this aspect in the discussion section.
Comment 5: The conclusion correctly correlates to the aim of your work, both mentioning the comparison of Neuro-Sjogren with CIDP, but this is not made clear in the title of your work. Therefore, I believe it to be necessary for you to either better justify this choice of comparison - is CCM already established as a diagnostic tool in CIDP? - or to change your title to better describe the scope of your work.
Response 5: We are aware that we have not sufficiently explained our reasons for this study design: We chose the CIDP group because although CCM is not yet an established diagnostic tool in CIDP patients, it is increasingly being used. In addition, the disease CIDP is an obvious comparator for patients with Sjögren’s syndrome and neuropathy because of the many similar clinical features.
Furthermore, our previous work in patients with CIDP showed a higher frequency of patients with cranial nerve affection in the subgroup with additional Sjögren’s syndrome compared to CIDP patients without Sjögren’s syndrome. Assessment of the trigeminal nerve as a cranial nerve is well feasible by CCM, which is why we aimed to focus on the characteristics of the trigeminal nerve in this study.
We have considered these aspects in the introduction.
Reviewer 3 Report
This study evaluates the corneal trigeminal nerve affection in patients with Sjögren’s syndrome(SS) associated neuropathy compared to patients with chronic inflammatory demyelinating polyneuropathy (CIDP) and searches for possible connections with dry-eye disease symptoms and signs. The authors conclude that no difference was found between the two neuropathy groups; findings indicate that CCM cannot distinguish between these two entities. The study has many methodologic problems. More specifically:
1. This Sjögren’s syndrome patient group is rather peculiar! It includes 42% males. The usual Sjögren’s syndrome patient cohorts have over 90% females.
2. Why were autoantibodies only to Ro antigen analyzed in the study and not to La antigen, which is more Sjögren’s syndrome specific?
3. The description of peripheral nerve affection in patients with Sjögren’s syndrome is poor! From what type of peripheral neuritis the SS patients were suffering from? Peripheral neuropathies in Sjögren’s syndrome: Peripheral neuropathy associated with sicca complex. Neurology Apr 1997, 48 (4) 855-892; DOI: 10.1212/WNL.48.4.855 and critical update on clinical features and pathogenetic mechanisms. Journal of Autoimmunity, (2012); 39(1-2), 27-33. doi:10.1016/j.jaut.2012.01.003
4. Did this SS patient have circulating monoclonal Immunoglobulins, light chains, or cryoglobulins?
5. Which type of autoantibodies the CIDP patient group had? Brain, Volume 140, Issue 7, July 2017, Pages 1851–1858
6. It would have been more relevant if the authors instead of using patients with chronic inflammatory demyelinating polyneuropathy, as disease controls, had included in the study a patient group with another autoimmune rheumatic disorder and peripheral nerve affection similar to one seen in patients with Sjögren’s syndrome.
Author Response
First of all, we would like to thank the kind reviewers for taking time to review our manuscript and providing a fair and critical feedback. We have addressed all points raised by the reviewers and provide a detailed point to point response.
Comment 1: This Sjögren’s syndrome patient group is rather peculiar! It includes 42% males. The usual Sjögren’s syndrome patient cohorts have over 90% females.
Response 1: The fact that the presented cohort includes a larger portion of male patients than other, mainly rheumatologic Sjögren’s syndrome cohorts without a focus on neurological involvement is a feature of neurological patients. In our previous work we found a similar shift in the male/female-ratio (Seeliger et al., 2019, 2021). A possible explanation from our point of view would be that patients with Sjögren’s syndrome and neurological involvement present with a different phenotype than patients with Sjögren’s syndrome but without neurological involvement. We have also experienced that other organ manifestations of Sjögren’s syndrome are rare in our neurologically oriented cohort. However, these observations are still the subject of ongoing investigation.
Comment 2: Why were autoantibodies only to Ro antigen analyzed in the study and not to La antigen, which is more Sjögren’s syndrome specific?
Response 2: We analyzed SSA(Ro)-antibodies and listed them here as they are part of the current classification criteria. This paper was not primarily focused on immunological parameters, so we had not listed the other antibodies. Following the suggestion of including information about the SSB(La)-antibody status, we now additionally analyzed the according laboratory results and integrated the results in new table 3.
Comment 3: The description of peripheral nerve affection in patients with Sjögren’s syndrome is poor! From what type of peripheral neuritis the SS patients were suffering from? Peripheral neuropathies in Sjögren’s syndrome: Peripheral neuropathy associated with sicca complex. Neurology Apr 1997, 48 (4) 855-892; DOI: 10.1212/WNL.48.4.855 and critical update on clinical features and pathogenetic mechanisms. Journal of Autoimmunity, (2012); 39(1-2), 27-33. doi:10.1016/j.jaut.2012.01.003
Response 3: We apologize that our clinical description of peripheral nerve impairment in our cohort of patients with Sjögren's syndrome was not detailed enough. We have therefore included more information in a new table 3.
Comment 4: Did this SS patient have circulating monoclonal Immunoglobulins, light chains, or cryoglobulins?
Response 4: As suggested, we expanded the analysis of serological parameters for patients with Sjögren’s syndrome: Cryoglobulins were not found in any patient and monoclonal immunoglobulin synthesis was evident in 5 patients (- which can occur in this entity (Bai et al., 2022). Information on free light chains (kappa and lambda) was available for 13/26 patients. We included this information in the new table 3.
Comment 5: Which type of autoantibodies the CIDP patient group had? Brain, Volume 140, Issue 7, July 2017, Pages 1851–1858
Response 5: We agree that information on the autoantibody status of CIDP patients would be interesting, as recent studies have been able to reflect the heterogeneity of this entity through serological biomarkers (Delmont et al., 2017). Unfortunately, use of these biomarkers has not yet been integrated into the daily routine of CIDP diagnosis, so information on specific autoantibodies is still not widely available. Therefore, retrospective integration of theses markers is not feasible for our study, although it represents a promising approach for future studies. Nevertheless, we have this information in the methods section.
Comment 6: It would have been more relevant if the authors instead of using patients with chronic inflammatory demyelinating polyneuropathy, as disease controls, had included in the study a patient group with another autoimmune rheumatic disorder and peripheral nerve affection similar to one seen in patients with Sjögren’s syndrome.
Response 6: It is true that the best disease control for a study is clinically similar but clearly definable and different from the entity in focus. In our previous work from 2021, we analyzed 154 patients who met the diagnostic criteria for CIDP and compared the clinical features of the subgroup with additional Sjögren’s syndrome (N=54) and without (N=100) (Seeliger et al., 2021). We found that the clinical phenotype of both patient groups was quite similar, except for a plausibly higher portion of female patients and cranial nerve impairment in the subgroup with additional Sjögren’s syndrome. Furthermore, both CIDP and neuropathy associated with Sjögren’s syndrome are autoinflammatory diseases that lead to peripheral nerve damage, although the pathomechanisms are likely to be different. For this reason, we deliberately selected CIDP patients as disease controls for our cohort of patients with Sjögren’s syndrome and neurological involvement.
References
Bai, Z., Hu, C., Zhong, J., & Dong, L. (2022). Prevalence and Risk Factors of Monoclonal Gammopathy in Patients with Autoimmune Inflammatory Rheumatic Disease: A Systematic Review and Meta-Analysis. Modern Rheumatology. https://doi.org/10.1093/MR/ROAC066
Delmont, E., Manso, C., Querol, L., Cortese, A., Berardinelli, A., Lozza, A., Belghazi, M., Malissart, P., Labauge, P., Taieb, G., Yuki, N., Illa, I., Attarian, S., & Devaux, J. J. (2017). Autoantibodies to nodal isoforms of neurofascin in chronic inflammatory demyelinating polyneuropathy. Brain, 140(7), 1851–1858. https://doi.org/10.1093/BRAIN/AWX124
Seeliger, T., Gingele, S., Bönig, L., Konen, F. F., Körner, S., Prenzler, N., Thiele, T., Ernst, D., Witte, T., Stangel, M., & Skripuletz, T. (2021). CIDP associated with Sjögren’s syndrome. Journal of Neurology, 1, 3. https://doi.org/10.1007/s00415-021-10459-z
Seeliger, T., Prenzler, N. K., Gingele, S., Seeliger, B., Körner, S., Thiele, T., Bönig, L., Sühs, K.-W., Witte, T., Stangel, M., & Skripuletz, T. (2019). Neuro-Sjögren: Peripheral neuropathy with limb weakness in Sjögren’s syndrome. Frontiers in Immunology, 10, 1600. https://doi.org/10.3389/fimmu.2019.01600
Round 2
Reviewer 3 Report
No further comments
Author Response
no further comments